# The properties of human disease mutations at protein interfaces

**Benjamin J. Livesey**, **Joseph A. Marsh** *

MRC Human Genetics Unit, Institute of Genetics & Cancer, University of Edinburgh, Edinburgh, United Kingdom

* joseph.marsh@ed.ac.uk

## Abstract

The assembly of proteins into complexes and their interactions with other biomolecules are often vital for their biological function. While it is known that mutations at protein interfaces have a high potential to be damaging and cause human genetic disease, there has been relatively little consideration for how this varies between different types of interfaces. Here we investigate the properties of human pathogenic and putatively benign missense variants at homomeric (isologous and heterologous), heteromeric, DNA, RNA and other ligand interfaces, and at different regions in proteins with respect to those interfaces. We find that different types of interfaces vary greatly in their propensity to be associated with pathogenic mutations, with homomeric heterologous and DNA interfaces being particularly enriched in disease. We also find that residues that do not directly participate in an interface, but are close in three-dimensional space, show a significant disease enrichment. Finally, we observe that mutations at different types of interfaces tend to have distinct property changes when undergoing amino acid substitutions associated with disease, and that this is linked to substantial variability in their identification by computational variant effect predictors.

## Author summary

Nearly all proteins interact with other molecules as part of their biological function. For example, proteins can interact with other copies of the same type of protein, with different proteins, with DNA, or with small ligand molecules. Many mutations at protein interfaces, the regions of proteins that interact with other molecules, are known to cause human genetic disease. In this study, we first investigate how different types of protein interfaces have different tendencies to be associated with disease. We also show that the closer a mutation is to an interface, the more likely it is to cause disease. Finally, we study how mutations at different types of interfaces tend to be associated with different changes in amino acid properties, which appears to influence our ability to computationally predict the effects of mutations. Ultimately, we hope that consideration of protein interface properties will eventually improve our ability to identify new disease-causing mutations.

**Data Availability Statement:** Supporting data for this manuscript has been deposited at https://doi.org/10.6084/m9.figshare.17040524.

**Funding:** JAM was supported by a Medical Research Council (https://mrc.ukri.org/) Career Development Award (MR/M02122X/1) and is a

Lister Institute (https://www.lister-institute.org.uk/) Research Prize Fellow. BJL was supported by a Medical Research Council Precision Medicine Doctoral Training Programme studentship. The funders had no role in study design, data collection and analysis, decision to publish, or preparation of the manuscript.

**Competing interests:** The authors have declared that no competing interests exist.

## Introduction

Single nucleotide variants (SNVs) are the most common type of human genetic variation [1]. While many SNVs in protein-coding regions of the genome are associated with human disease, the vast majority have no noticeable clinical impact [2]. We have still seen only a fraction of the possible coding SNVs in genetic sequencing studies and most of these remain completely unannotated. Distinguishing novel disease-causing SNVs from those that are benign is therefore a major ongoing challenge for the field of bioinformatics.

Protein misfolding and destabilisation have long been held as primary mechanisms by which mutations cause disease [3], and it is well established that pathogenic missense mutations are enriched within interior residues of proteins relative to the surface [4–6]. In the last decade, there has also been increasing recognition of the importance of protein interfaces as hubs for disease-associated mutants [7–9]. A mutation at an interface can act to destabilise the interface, disrupt the binding of the partner, increase, reduce or alter the affinity of the partner or stabilise the interface [10], an association which has been shown to be highly associated with disease states [11]. Experimental work strongly supports the idea that many pathogenic alleles do not destabilise protein folding but instead perturb protein interactions [12].

The separation of interface residues into core and rim regions can be useful for understanding the mechanisms of pathogenesis at protein interfaces. Core regions are those buried within the interface and contain the highest enrichment of pathogenic variants [11,13,14]. Rim regions are located around the edge of the interface and are more exposed to solvent. Core and rim residues show differences in both physiochemical properties and conservation [11,15]. There is also evidence that rim residue mutations may be more impactful in smaller complexes [16].

It was noted some time ago, that mutations in specific "hot spot" residues within interfaces can greatly alter the free energy of binding [17] through disruption of non-covalent interactions [18]. It has since been shown that mutations of these hot spot residues are enriched for disease mutations relative to the rest of the interface [11,19,20]. Hot spots are preferentially found within the interface core, although those located in the rim remain enriched for pathogenic variants [11].

Interface mutations also appear to be particularly important in the context of cancer [21–23]. Those proteins impacted tend to be hub proteins within the human interactome. In p53 and other proteins, it was shown that patient survival was correlated to the specific interactions that were perturbed [21,24]. Mutations in interfaces allow for specific interactions to be abolished and altered, while the protein retains some activity [25]. This 'edgetic' perturbation allows cancer to re-wire the normal interactome with less risk of full collapse that may result from full protein destabilisation. A similar pattern of interaction destabilising mutants has also been observed for developmental disorders such as autism [26].

To improve the prediction of protein variant effects, it is important to first better understand the molecular mechanisms underlying damaging mutations and how these are related to amino acid properties. Previous studies have shown that tryptophan, tyrosine and arginine mutations in interfaces are overrepresented among disease-causing mutations [13,17]. Different amino acid substitutions also cause disease at different rates depending on where they occur in the protein or interface [11]. Drastic changes in physicochemical properties at interfaces were observed in cancer-associated mutations [22]. However, other studies have demonstrated that amino acid conservation is often a larger factor in predicting pathogenic mutations than the change in properties [27]. As with the protein interior residues, conservation remains an important factor for understanding the pathogenicity of mutations at protein interfaces [11,15].

One aspect that has not received much consideration in previous studies of protein interfaces is the nature and orientation of the interactors. Protein-protein interfaces are most easily divided by considering homomeric and heteromeric interfaces separately. Homomers can be further divided into isologous and heterologous interactors. Proteins can also interact with other biomolecules, such as DNA and RNA, and small-molecule ligands. In this study, we have taken interface type into consideration, in an attempt to better understand how the structural and sequence properties of different types of human protein interfaces are related to the effects of mutations. We investigate the enrichment of pathogenic mutations within different regions of proteins and different types of interfaces. We also investigate the property changes caused by mutations, and how these differ between protein regions. Finally, we show how these findings have implications for the prediction of variant effects at protein interfaces.

## Results and discussion

### Enrichment of disease mutations at different types of protein interfaces

To represent pathogenic variants in our study, we used missense variants from the ClinVar [28] database, including only those annotated as 'pathogenic' or 'likely pathogenic'. We also used missense variants observed in the human population from gnomAD v2.1 [29], excluding those also present in the ClinVar dataset. We refer to these variants as 'putatively benign', as while gnomAD excludes individuals with severe paediatric disease, recessive and low-penetrance variants are certainly present in the dataset to some extent. We considered only variants that could be mapped to residues in published PDB structures; this resulted in a total of 495,076 putatively benign missense variants in 4033 genes, and 19,175 (likely) pathogenic missense variants in 1754 genes.

First, we looked at how the locations of missense variants within protein, and protein complex structures are related to enrichment in disease. We used solvent accessible surface area (SASA) to classify residues in PDB structures of human proteins as surface, interior or interface (Fig 1A). Surface residues have 25% or more of their surface area [30] exposed to solvent, while interior residues have less than this. Interface residues undergo a change in SASA between the monomeric and complexed forms of the structure, meaning they participate directly in the interface.

By calculating the odds ratio of pathogenic to putatively benign variants at each location, we find that mutations on the protein surface are highly depleted of pathogenic variants (Fig 1B) (0.31 times, p<$2.225\times10^{-308}$), while interior positions are enriched (1.50 times, p = $1.06\times10^{-153}$). This interior enrichment differs somewhat from a previous estimate (2.66 times), but that study used different mutation datasets and a stricter definition of buried residues (<1% exposed surface area) [31]. Other studies that used a similar definition of interior residues to us obtained similar enrichment values [32]. Interestingly, we find an even greater enrichment of disease mutations within protein interfaces of 1.61 times (p = $1.83\times10^{-225}$). Another study [11], found that disease enrichment in interface residues was intermediate to interior and surface residues. This study also used a stricter definition for interior residues (7% SASA cutoff), as well as a definition for interface residues based on residue proximity rather than SASA changes, potentially accounting for the difference.

Next, we split the interface residues into three regions–core, rim and support–as previously defined [33]. Core interface residues are on the surface of the unbound subunit (≥25% of their surface area exposed in the monomer), but are buried in the bound complex, suggesting that they are likely to be crucial for the interaction. Support residues are buried in the unbound subunit (<25% of surface area exposed), suggesting that their main role may be in stabilising intramolecular structure, but that they also participate in the interaction. Rim residues are on

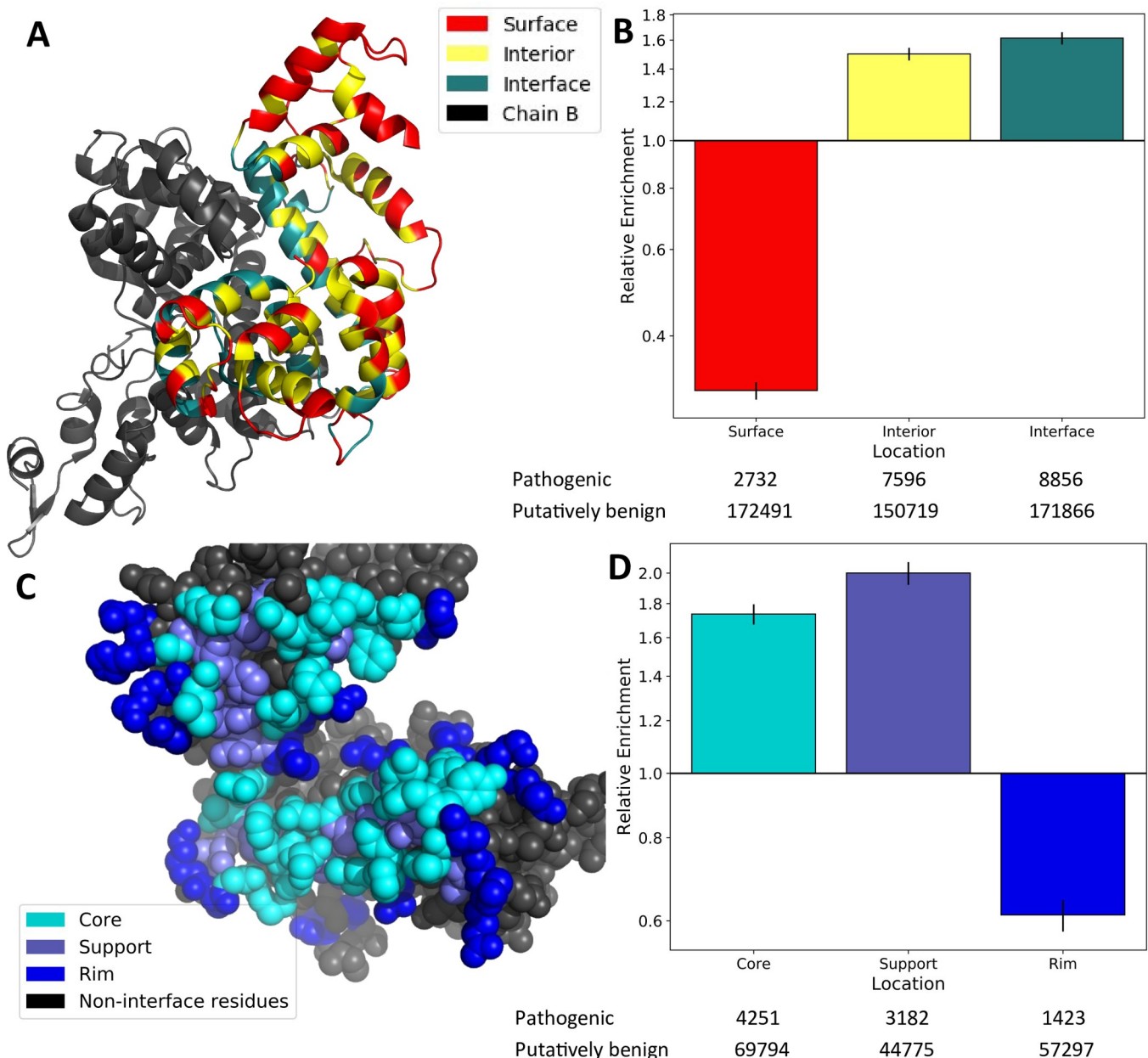

**Fig 1. The anatomy of protein interfaces and the enrichment of pathogenic mutations.** A) Ribbon representation of the structure of a homodimeric mitochondrial aspartate/glutamate carrier protein [78] (PDB ID: 4P60), highlighting surface, interior and interface residues. B) Relative enrichment (odds ratio) of pathogenic mutations within protein surface, interior and interface locations. C) Sphere representation of the 4P60 dimer homomeric interface on one chain, highlighting core, support and rim residues. D) Relative enrichment of pathogenic mutations within interface core, support and rim regions. All error bars represent 95% confidence intervals. Numbers of pathogenic and putatively benign mutations within each location dataset are shown below the plots. Detailed relative enrichment data is available in S1 Table.

the surface of both the monomer and full complex (≥25% surface area exposed in both), suggesting that they may be less important to both protein stability and the interaction. An example of these classifications is shown in Fig 1C.

The distribution of pathogenic variants within the different interface regions is not uniform (Fig 1D). Both support and core residues show a strong enrichment in pathogenic variants (2.00 times and 1.74 times respectively). In contrast, rim residues resemble surface residues in

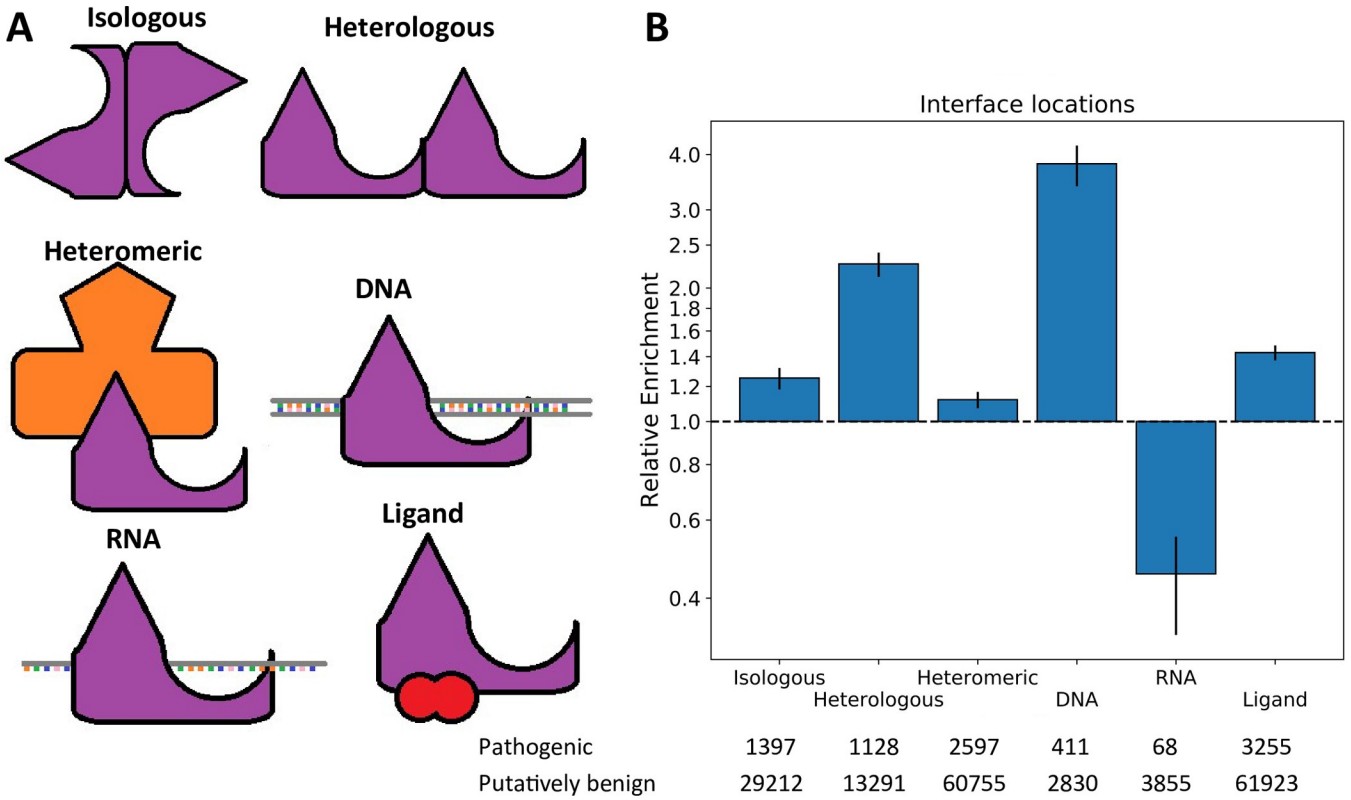

**Fig 2. Comparison of the enrichment of pathogenic mutations at different protein interfaces.** A) Demonstration of each protein interface type. B) Relative enrichment of pathogenic mutations within each interface type. Error bars represent 95% confidence intervals. Numbers of pathogenic and putatively benign mutations within each location dataset are shown. Detailed relative enrichment data is available in S1 Table.

that they show a depletion of pathogenic mutants (0.61 times, p = 1.68x10$^{-78}$). These results largely agree with previous studies of interface mutations where variants buried within protein interfaces demonstrate a greater tendency to cause disease than those on the interface rim [8,11]. Due to the fact that rim residues are most similar to surface residues in terms of their lack of disease enrichment, for subsequent analyses, we have grouped rim residues with other surface residues rather than with interface residues.

While previous work has clearly demonstrated the importance of protein interfaces for understanding disease mutations, there has been relatively little consideration of the different types of protein interfaces. Therefore, we further classified protein interfaces based upon the types of interactions they make (Fig 2A). Homomeric interfaces, formed by interaction between two copies of the same protein subunit, can be split up into those that are isologous (i.e. head-to-head or symmetric, involving the same surface patch on each subunit), and those that are heterologous (i.e. head-to-tail or asymmetric, involving different surface patches on each subunit) [34]. We also separately considered heteromeric interfaces (formed between two distinct protein subunits), as well as DNA, RNA and other ligand-binding interfaces.

Again, we calculated the odds ratios of pathogenic to benign variants for each interface type (Fig 2B). Strikingly, DNA interfaces showed the greatest enrichment of pathogenic variants (3.81 times, p = 8.54x10$^{-102}$), which we suspect may be at least partially due to the highly damaging effects of transcription factors with off-target effects [35,36]. We also observed that heterologous interfaces were more disease-enriched (2.26 times, p = 1.26x10$^{-118}$) than either isologous interfaces (1.25 times, p = 1.29x10$^{-14}$) or heteromeric interfaces (1.12 times,

p = 2.15x10$^{-7}$). Surprisingly, RNA interface residues were depleted in disease mutations (0.45 times, p = 2.37x10$^{-13}$). However, after excluding ribosomal proteins from the analysis, the enrichment becomes 1.04 (p = 0.71). One possible explanation for this is that ribosomal proteins are so highly conserved that many impactful variants are simply not observed in the population due to embryonic lethality.

Ligand interfaces also show a significant enrichment in pathogenic mutations (1.43 times, p = 1.21x10$^{-68}$). However, our definition of ligand-binding interfaces is based upon the presence of small molecule ligands present in PDB structures (excluding water). As such, both biologically relevant ligands and non-relevant crystallisation artefacts are present in our dataset. We used FireDB [37] to categorise ligands at these interfaces as 'cognate' or 'non-cognate' and investigated the enrichment of pathogenic variants within each category (Fig A in S1 Text). Cognate ligand interfaces are the most highly enriched for disease variants (2.04 times, p = 6.39x10$^{-76}$), although there is still a significant enrichment in non-cognate interfaces (1.24 times, p = 4.33x10$^{-20}$). While these substances are not natural biological ligands to the protein, they often include drug molecules that bind at conserved sites, thus potentially explaining this enrichment.

As noted earlier, the gnomAD dataset is imperfect as a representative of truly benign variation, and will contain some small proportion of damaging variants. One possible way to deal with this is to filter the gnomAD variants on the basis of allele frequency, and consider only those present in ≥1% of individuals. Unfortunately, this dramatically reduces the size of the dataset, from almost 500,000, to 2184. Another option is to only include those that have been annotated as 'benign' or 'likely benign' in ClinVar, leaving 4355 variants. We re-ran our enrichment analyses using different subsets of the gnomAD variants: common (≥1%) variants, rare (<1%) variants, and (likely) benign with ClinVar annotations (Fig B in S1 Text). Using common gnomAD variants tends to produce slightly stronger enrichment trends, suggesting that they are more representative of truly benign variants, as expected. Overall, however, we observe remarkably similar patterns regardless of the 'benign' subset used for the comparison. This suggests that rare damaging variants present in the full gnomAD data have relatively little impact on our analyses. Therefore, to maximise the power of our analyses, we utilise the full gnomAD dataset for the remainder of the study.

## Interface enrichment *vs* complex symmetry, biological function and interface size

The fact that the two different types of homomeric interfaces, isologous and heterologous, show very different enrichments in pathogenic mutations is very interesting. Homomeric interface type is closely related to complex symmetry, with twofold symmetric dimers ($C_2$) having isologous interfaces, complexes with higher order cyclic symmetry ($C_{n;\ n>2}$) having heterologous interfaces, dihedral complexes ($D_n$) having isologous and sometimes heterologous interfaces, and complexes with helical ($H$) symmetry having exclusively heterologous interfaces [38]. It is possible that this overrepresentation of pathogenic mutations in heterologous interfaces is driven by a disease association with complex symmetry, as has previously been suggested [39]. We therefore investigated the enrichment of disease mutations within homomers belonging to different symmetry groups (Fig 3A). Our results indicate that all homomeric complexes show greater enrichment in disease mutations compared to monomers that do not self-associate. Within the homomers, only the symmetric dimers show a slight overall depletion in pathogenic mutants, while cyclic, dihedral and helical complexes all show an enrichment. One possible explanation for this could be that larger complexes with more subunits have relatively more interface residues and relatively fewer surface residues. However, we

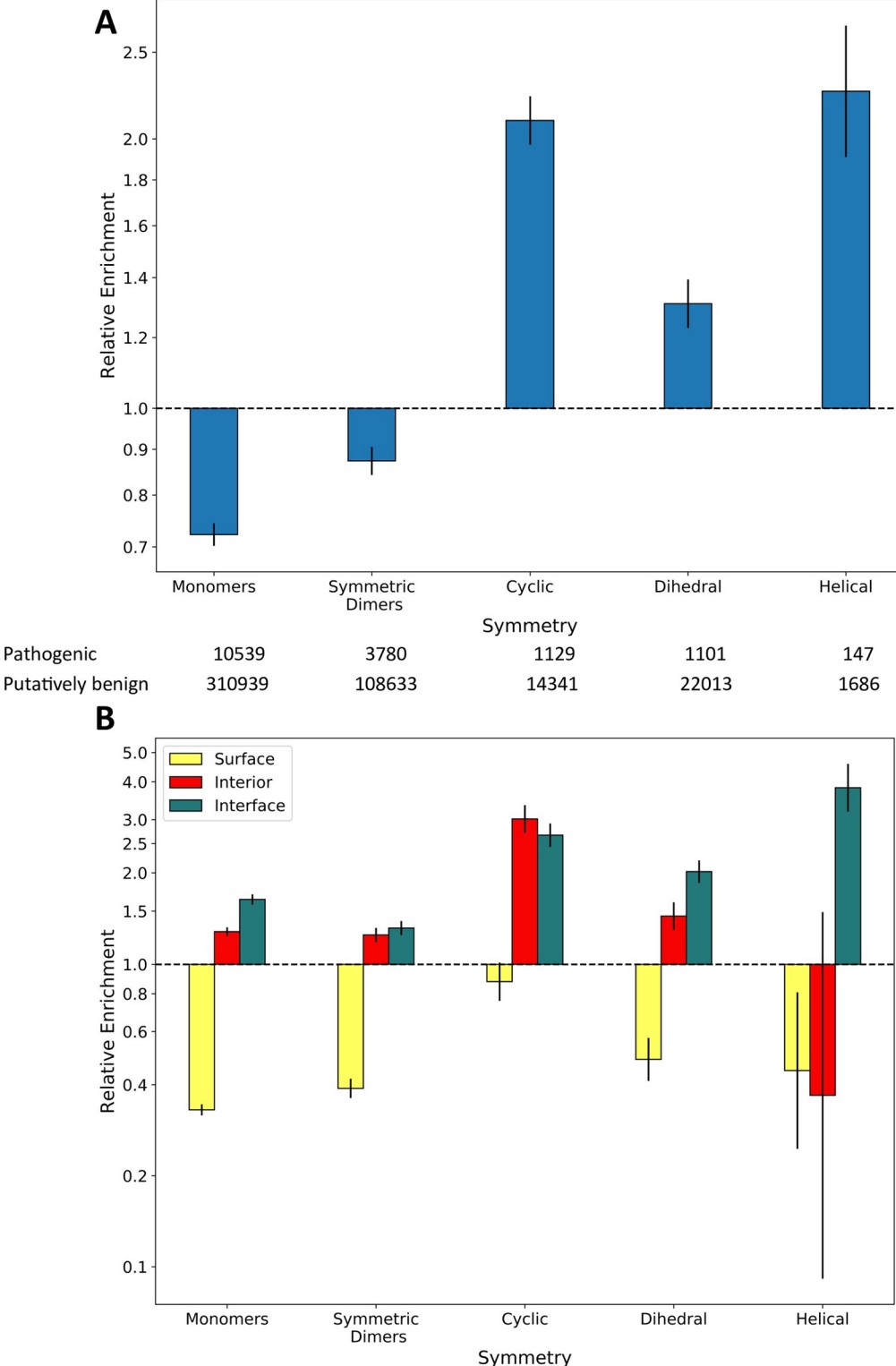

| | Monomers | Symmetric Dimers | Cyclic | Dihedral | Helical |
|---|---|---|---|---|---|
| Pathogenic | 10539 | 3780 | 1129 | 1101 | 147 |
| Putatively benign | 310939 | 108633 | 14341 | 22013 | 1686 |

**Fig 3. The enrichment of pathogenic mutations within protein complexes of different symmetries.** A) Relative enrichment of pathogenic mutations within proteins of each symmetry type. Symmetric dimers ($C_2$) are considered separately from higher-order cyclic ($C_{n;\ n>2}$) complexes. Complexes with cubic symmetries are not included due to lack of representation in the data. B) Enrichment of pathogenic mutations in the surface, interior and interface of each symmetry type relative to the full dataset. All error bars represent 95% confidence intervals. Detailed relative enrichment data is available in S1 Table.

observe that interior and interface residues in cyclic and dihedral complexes show a greater disease enrichment than in monomers and symmetric dimers (Fig 3B). Although interior residues in helical complexes are actually depleted in disease mutations, it is important to note the small number of helical complexes in our dataset and the corresponding large range covered by the confidence interval.

The enrichment of pathogenic mutations may also be related to functional associations of the different symmetry groups, *e.g.* dihedral complexes are often enzymes while cyclic complexes include many transmembrane channels [40]. To investigate whether a high proportion of membrane proteins with heterologous interfaces were biasing our analysis, we repeated the calculations while excluding all proteins tagged with the keyword 'membrane' (KW-0472) in UniProt (Fig C in S1 Text). Heterologous interfaces remained more enriched for pathogenic variants (1.87 times, $p = 6.93 \times 10^{-33}$) than either isologous or heteromeric interfaces. We also further investigated how different biological functions might be influencing our results by calculating the enrichment of pathogenic mutations within different locations and interface types, when controlling for three common functional annotations from UniProt: 'catalytic activity', 'transcriptional regulator' and 'transporter activity' (Fig D in S1 Text). Overall, we observe similar patterns within the three groups compared to the full set of human proteins, suggesting that our results are not strongly confounded by disease association with different biological processes.

Another factor that may influence how pathogenic mutations are distributed is the size of the interface. Previous work has shown that transient complexes are more likely to have smaller interface areas, while more permanent complexes have larger interfaces [41,42]. Similarly, larger interfaces will tend to form earlier during protein complex assembly, while smaller interfaces will form later [43–45]. We calculated the sizes of all interfaces in our dataset. We then split our variants into three equally sized groups for each interface type, based on whether they occur in small, medium or large interfaces. Our results indicate a strong correspondence between size and disease enrichment for certain interface types (Fig E in S1 Text). For both types of homomeric interfaces, there is a highly significant tendency for pathogenic mutations to be enriched in larger interfaces. However, for both heteromeric and ligand interfaces, there is no significant trend. This may reflect the fact that smaller homomeric interfaces are much more likely to be crystal-packing artefacts and thus not biologically important [46]. In contrast, a given heteromeric interface is much more likely to be biologically important, regardless of whether it is small or large. Similarly, the likelihood of a ligand interaction being biologically important may be largely independent of ligand size, given that many important ligands like metal ions will form only very small interfaces. Intriguingly, disease mutations are enriched in small DNA and RNA interfaces, although the trend is only significant for DNA. Possibly this may be due to many small DNA-binding domains from transcription factors being highly enriched in disease-causing mutations, compared to other proteins that interact with DNA.

Finally, we also investigated other factors that could potentially influence the enrichment trends we have observed. In Fig F in S1 Text, we calculate the disease enrichment at different locations for proteins of different lengths, and for proteins with different numbers of variants in our dataset. We find similar patterns of enrichment within and outside of these subsets, showing that these properties cannot explain our observations.

## Enrichment of disease mutations in proximity to interfaces

Our observations show that the support residues of protein interfaces, which are most buried intramolecularly within the protein structure, clearly have a greater enrichment of disease mutations than the interface core, or interior residues. This is most likely due to the combined

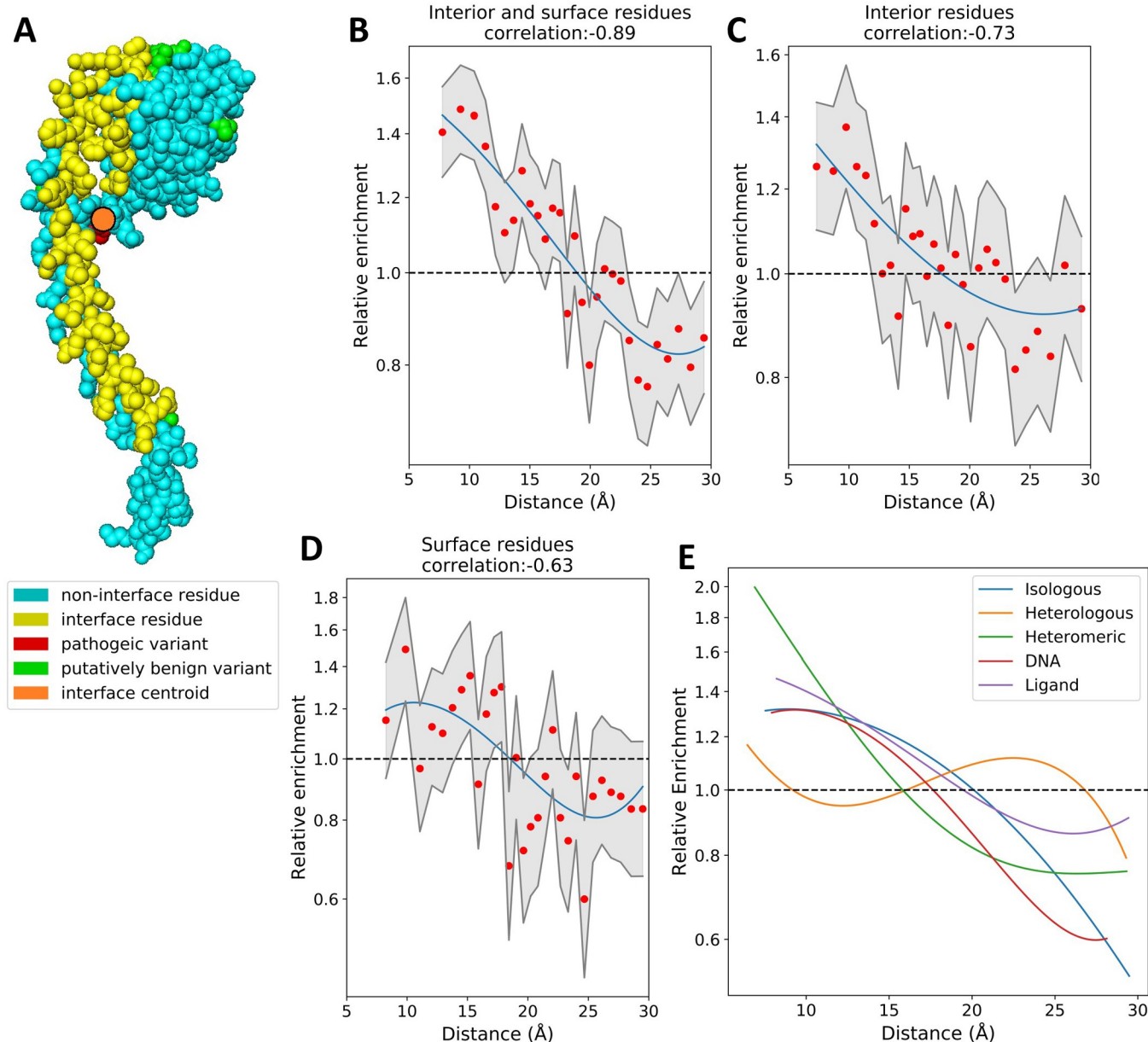

**Fig 4. The relationship between interface proximity and disease enrichment.** A) Sphere representation of a protein with interface residues highlighted in yellow. The interface centroid is an orange circle. Putatively benign mutations present in gnomAD are shown in green, while pathogenic ClinVar mutations are in red. B-D) Odds ratio of disease mutations at increasing distance from the nearest interface centroid. Confidence intervals (95%) are shown in grey, a univariate spline (blue) has been fit to the data. Charts are shown for surface and interior residues together and individually. E) Overlaid splines fit to odds ratio data of the different protein interface types (taking interior and surface residues together).

effect of support mutations destabilising both folding and intermolecular interactions. We therefore hypothesised that other interior residues in proximity to the interface would show similar levels of disease enrichment, despite not participating in the interface directly.

To address this, we calculated the proximity of residues to the nearest interface and compared this to their disease enrichment. We utilized the interface centroid, which is defined as the mean of the X, Y and Z coordinates of all residue alpha-carbons in the interface (Fig 4A). We generated 30 bins containing identical numbers of variants at increasing distance from the

nearest interface centroid, and identified a strong correlation (-0.89 Spearman's ρ) between this distance and the enrichment of disease mutations within the bins (Fig 4B). Notably, a greater correlation was found (-0.73) when we consider interior residues alone (Fig 4C), than when we consider surface residues alone (-0.63) (Fig 4D). Interior residues in close proximity to the nearest interface centroid can be considered as pseudo-support residues, directly influencing the buried residues in the interface. Our results suggest that this effect does not terminate directly adjacent to the interface and instead propagates further through the structure. The effect of interface proximity on disease risk does not appear to influence the protein surface as much as the interior, possibly due to the lower density of intramolecular contacts available for propagating mutation effects.

We believe that this effect is primarily caused by the propagation of destabilising effects from a variant in a distance-dependent manner. The potential for interface disruption is highest in close proximity to the variant, and drops off at greater distances. Allosteric residues may contribute to this effect, by being mutated directly, or by assisting the propagation of the effect to the interface. Allosteric residues are known to be highly conserved and have been used to explain the effects of several uncharacterised pathogenic mutants [47]. Recently, two allosteric F-actin mutants, which are distal to the myosin interface, were found to impact activity at that interface [48]; this is similar to the overall pattern we observe.

In addition to the interface centroid, we also investigated the minimum distance between a mutation and any interface residue, and the mean distance to all residues in the interface as interface proximity metrics (Fig G in S1 Text). A comparable pattern is found regardless of the metric used; however, the correlations are greatest when using centroid distance. Centroid distance is closely related to average distance, with a Spearman's correlations of 0.93 between the two metrics. Minimum distance is less closely related with correlations of 0.65 to centroid distance and 0.59 to average distance.

We observe a slight difference in the propagation of disease enrichment in different interface types (Fig 4E). Both isologous and heterologous homomeric interfaces display a greater enrichment of disease mutations between 15 and 25 Å from the interface centroid than heteromeric interfaces do. We also find that disease enrichment in DNA interfaces drops off more rapidly than in protein interfaces. We speculate that this could be due to specificity-altering mutations in DNA interfaces needing to be closer to the interface surface, while those further back either completely abolish the interaction or have no effect. Ligand interfaces show a shallower curve than any of the other interface types, although it crosses the neutral-enrichment point at roughly the same location, perhaps reflecting their smaller interfaces, or the fact that many ligands in protein structures are not biologically important (*e.g.* crystallisation artefacts).

## Properties of amino acid substitutions at interfaces

Given that protein interfaces tend to involve different types of amino acids than protein interiors or surfaces [33,49], we wondered whether this could be reflected in differential propensities for pathogenic amino acid substitutions. We therefore calculated the residue-level disease enrichment for both the wild-type and mutant amino acids for different locations (Fig 5 and Fig H in S1 Text).

Across interior, surface and interface locations, the enrichments show broad similarities, with mutations from arginine, tryptophan, cysteine and glycine wild-type residues most likely to result in a pathogenic outcome, while mutations to tryptophan and proline were the most likely to be pathogenic. This largely agrees with previous findings [50]. In surface residues, mutations from cysteine were more damaging than at interior locations. This is in-line with previous work, showing that thiol groups on protein surfaces play a role in regulating the

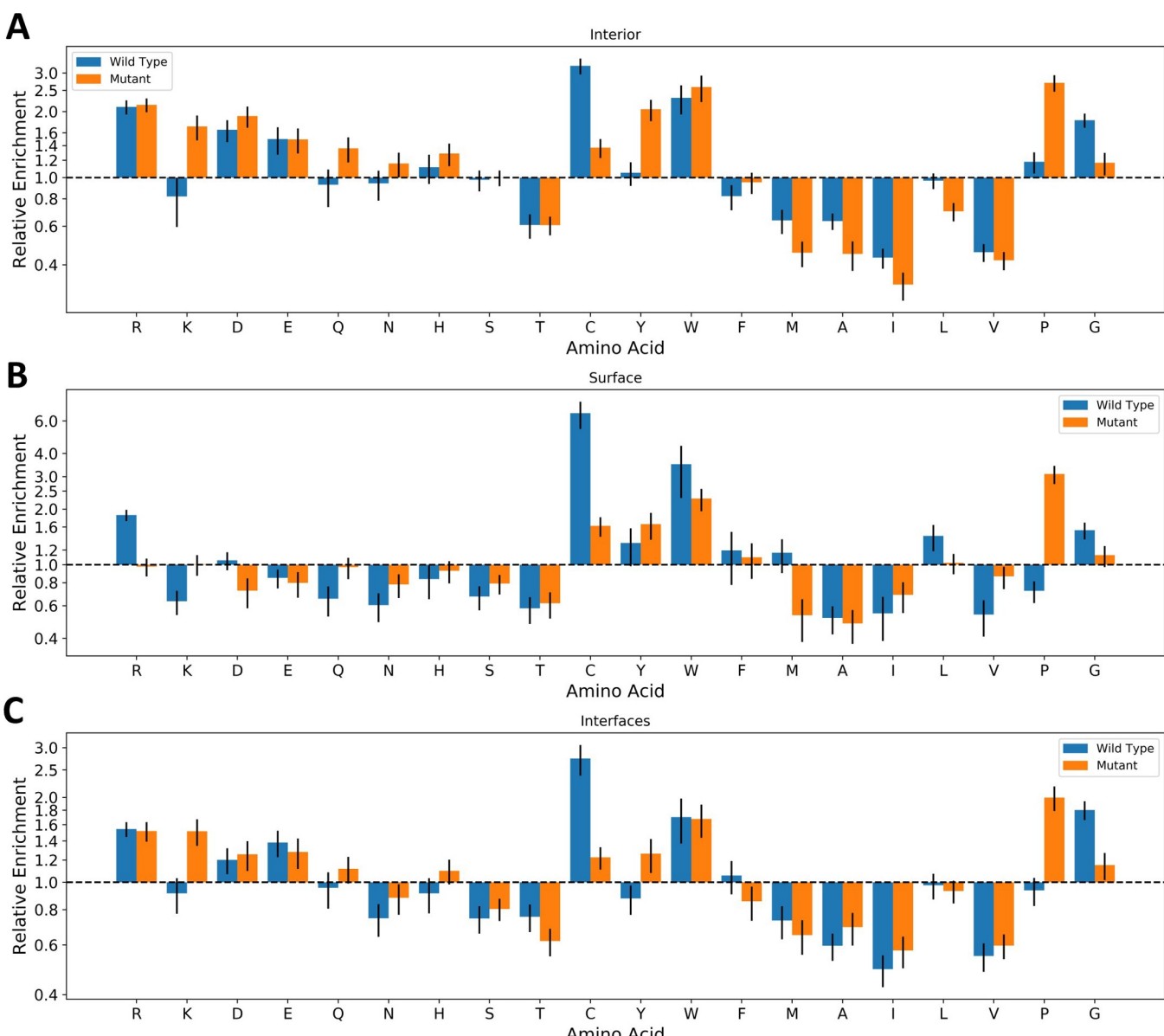

**Fig 5. Enrichment of pathogenic variants at specific residues at different locations.** The odds ratio of pathogenic mutations associated with specific amino acid mutations at A) the protein interior, B) the protein surface and C) protein interfaces. Blue bars represent mutations *from* a specific amino acid, while orange bars represent mutations *to* a specific amino acid. All error bars represent 95% confidence intervals. Detailed relative enrichment data is available in S1 Table.

cellular redox environment [51,52]. Substitution of charged residues clearly results in more pathogenic mutants on the interior, while hydrophobic substitutions are more damaging on the surface with the exceptions of tryptophan and glycine. Aspartate mutations are differentially enriched (enriched in pathogenic mutants on the interior, but depleted on the surface), and clusters with other charged amino acids which are either less differentially enriched or neither enriched nor depleted at surface locations.

Given that interface core and support residues are buried upon complex formation, we naturally can expect the mutational profile of interfaces to be similar to that of the protein interior. Comparing the relative enrichment of disease mutations in interior and interface

mutations, we find that substitutions involving charged and polar amino acids as either the wild type or the mutant are more likely to be damaging for the interior. In contrast, small hydrophobic amino acid substitutions are more likely to be damaging within interfaces. These results imply that disruption of hydrophobic interactions is an important mechanism for pathogenesis in interfaces.

Mutations from wild-type cysteine residues and to mutant proline residues both stood out in this analysis as having large enrichments of pathogenic variants, regardless of location. To investigate this further we analysed the enrichment of pathogenic mutations for each amino acid that could results from an SNV at a cysteine residue, and each SNV that could result in a substitution to proline (Fig I in S1 Text). Our results show that mutations from cysteine are roughly equally damaging when they occur at surface residues. However, in interior and interface locations, mutations to bulky (phenylalanine, tyrosine, tryptophan) or charged (arginine) amino acids were more damaging than mutations to glycine or serine. The damaging effects of proline substitutions were less location-dependent, but we observed that mutations from leucine and arginine tended to result in more pathogenic variants than other amino acids.

While investigating the mutational profiles of the different interface types, we found that our analysis of the residue mutations lacked power due to sub-setting of the data across the large number of possible amino acid substitutions. To supplement this analysis, we downloaded 547 amino acid physical, biochemical and statistical properties from AAindex [53]. By using amino acid properties, all mutations at a location can now contribute towards the results of our analysis. For each property, we calculated the difference between the wild type and mutant amino acids in our data sets. We then calculated the p-value and effect size of the mean difference between the datasets. Initially we found only small effect sizes ($<0.3$ Hedges' g), however, taking the absolute difference between wild type and the mutant greatly increased the effect. This implies that the scale of deviation from the wild-type property is more important than the direction of the deviation for most pathogenic mutations. The top three properties by effect size per location are shown in Table 1. The full data is available in S2 Table.

There are several properties with high effect sizes that distinguish pathogenic from benign mutants at multiple locations. These include mutants that affect hydrophobicity, hydrophilicity and related properties such as percentage of buried residues and accessible surface area. Since an altered hydrophobic profile can influence protein folding, this is unsurprising. *Average non-bonded energy per atom*, the property with the largest effect size over all locations, is the calculated contribution of each residue to the non-bonded energy of proteins. This property is also related to hydrophobicity, as hydrophobic interactions contribute greatly towards the total non-bonded energy [54]. *Principle property value z3* [55] is also among the top properties that distinguish pathogenic from putatively benign mutations at interior and surface locations as well as heteromeric interfaces. This is essentially a principal component derived from multiple other properties.

Most of the properties with the highest effect sizes are either related to hydrophobicity or to free energy changes, with the exception of two locations: homomeric heterologous interfaces and DNA interfaces. In these locations, properties related to secondary structure, helices specifically, also have moderate effect sizes. This hints at potentially different mechanisms of pathogenesis at each of these sites via the perturbation of secondary structure.

The change in effect size ranking for each property was calculated for every pair of locations. We visualised the difference in the property changes in Fig 6. These graphs plot change in rank against change in effect size for all AAindex properties at a pair of locations. A wider distribution indicates greater differences between the property's changes occurring in pathogenic vs benign mutations at the two locations. Properties that are highly skewed towards one corner of the plot show a large pathogenic/benign property difference at one location, but a

**Table 1. Top three property changes by effect size at different protein locations.**

| Location | Property | Average difference in gnomAD | Average difference in ClinVar | P-value | Effect size |
|---|---|---|---|---|---|
| **All locations** | Average non-bonded energy per atom | 0.1743 | 0.2297 | <2.225E-308 | 0.3824 |
| | Percentage of buried residues | 20.0986 | 26.2642 | <2.225E-308 | 0.3782 |
| | Hydrophility value | 1.6181 | 2.1795 | <2.225E-308 | 0.3710 |
| **Interior** | Hydrophilicity value | 1.3205 | 2.0982 | <2.225E-308 | 0.5792 |
| | Hydrophobic parameter | 1.2937 | 2.0303 | <2.225E-308 | 0.5646 |
| | Principle property value z3 | 1.6334 | 2.4645 | <2.225E-308 | 0.5524 |
| **Surface** | Hydrophobic parameter pi | 0.8531 | 1.1354 | 6.14E-133 | 0.3847 |
| | Principle property value z3 | 2.1558 | 2.7798 | 2.53E-129 | 0.3794 |
| | Side chain hydropathy, corrected for solvation | 1.9225 | 2.5670 | 4.69E-127 | 0.3760 |
| **Interfaces** | Apparent partition energies calculated from Janin index | 0.6465 | 0.8385 | 5.94E-190 | 0.3525 |
| | Average non-bonded energy per atom | 0.1748 | 0.2264 | 3.88E-189 | 0.3517 |
| | Hydrophilicity value | 1.6301 | 2.1618 | 3.80E-188 | 0.3508 |
| **Isologous Interfaces** | Transfer free energy | 0.6610 | 0.9084 | 9.62E-44 | 0.4293 |
| | Apparent partition energies calculated from Janin index | 0.6434 | 0.8763 | 1.46E-43 | 0.4284 |
| | Average accessible surface area | 22.1445 | 30.9438 | 3.21E-42 | 0.4214 |
| **Heterologous Interfaces** | Normalized positional residue frequency at helix termini C' | 0.4753 | 0.7315 | 5.83E-37 | 0.4343 |
| | Side chain angle theta(AAR) | 19.9685 | 36.1640 | 6.45E-34 | 0.4149 |
| | Normalized frequency of coil | 0.4166 | 0.6041 | 2.73E-33 | 0.4108 |
| **Heteromeric Interfaces** | Principle property value z3 | 1.9069 | 2.5636 | 1.36E-72 | 0.4014 |
| | Average non-bonded energy per atom | 0.1720 | 0.2293 | 5.64E-71 | 0.3968 |
| | Apparent partition energies calculated from Janin index | 0.6333 | 0.8459 | 7.29E-71 | 0.3965 |
| **DNA interfaces** | Dependence of partition coefficient on ionic strength | 0.0901 | 0.1288 | 3.37E-12 | 0.4153 |
| | Normalized frequency of N-terminal helix | 0.4458 | 0.6259 | 9.71E-11 | 0.3857 |
| | Helix termination parameter at position j+1 | 0.3895 | 0.5264 | 3.78E-09 | 0.3510 |
| **RNA interfaces** | Free energy change of alpha(Ri) to alpha(Rh) | 0.3104 | 0.4626 | 4.54E-05 | 0.5604 |
| | Partial specific volume | 0.0855 | 0.1148 | 3.18E-04 | 0.4945 |
| | Hydrophobicity coefficient in RP-HPLC, C18 with 0.1% TFA/MeCN/H2O | 1.5660 | 2.0809 | 3.64E-04 | 0.4897 |
| **Ligand interfaces** | Average non-bonded energy per atom | 0.1766 | 0.2235 | 1.29E-60 | 0.3193 |
| | Percentage of exposed residues | 19.8774 | 25.2315 | 3.13E-58 | 0.3127 |
| | Apparent partition energies calculated from Janin index | 0.6498 | 0.8204 | 4.74E-58 | 0.3122 |

small one at the other, indicating that these properties may be considerably more important for residues at one location than the other. Also marked on the plots are those properties with the greatest effect size at each location. The presence of these properties at the centre of the plot indicates that these properties are equally important at the two locations, and if they are highly skewed, then the opposite.

S3 Table contains summary statistics for the distributions plotted in Fig 6. The sum of all absolute effect size deltas and the sum of all absolute rank deltas indicate the amount of deviation away from 0 (no difference in property changes between the two locations) in a distribution on the y and x axes, respectively. The range of effect sizes and range of rank deltas are calculated by deducting the minimum value from the maximum for each axis, this gives some

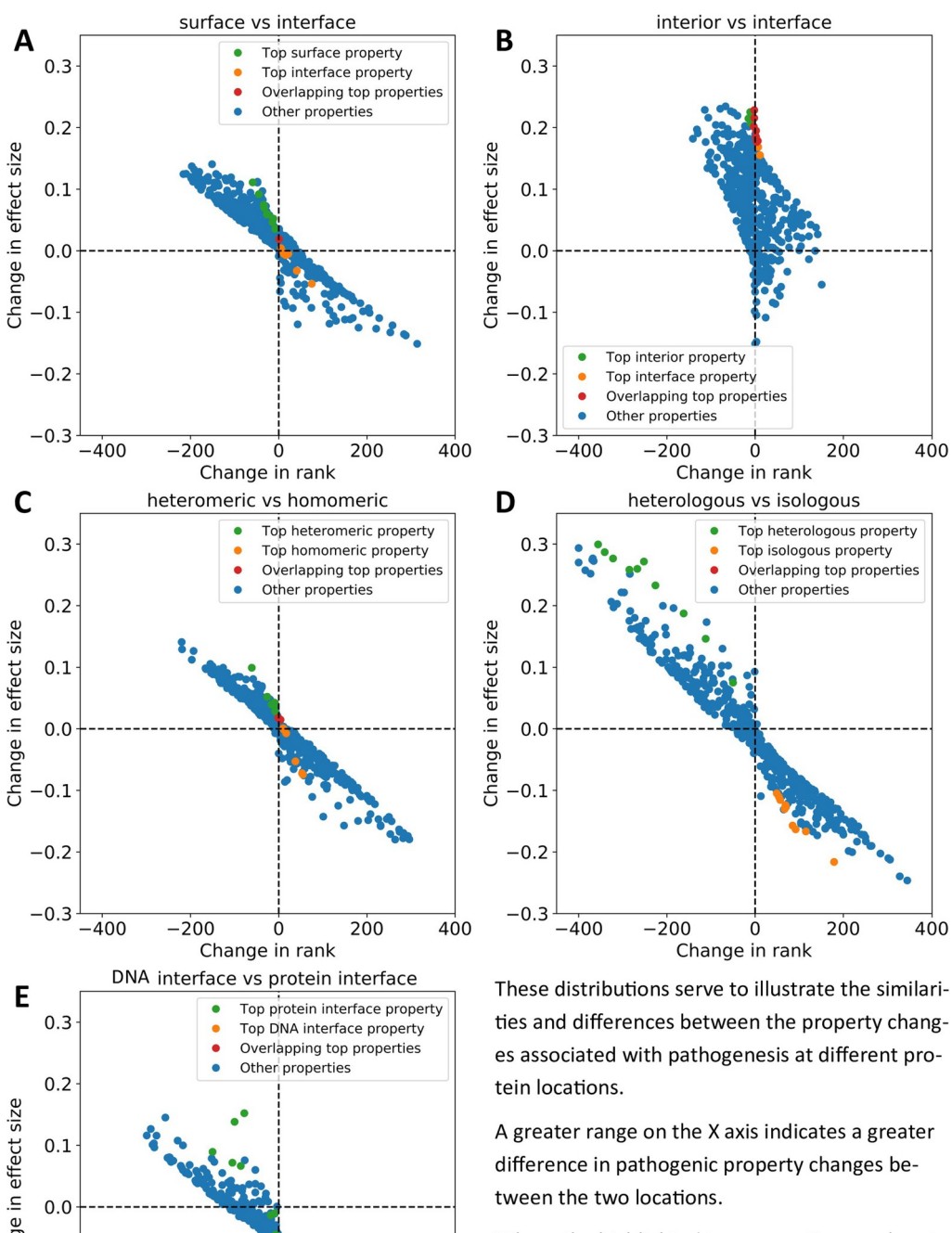

These distributions serve to illustrate the similarities and differences between the property changes associated with pathogenesis at different protein locations.

A greater range on the X axis indicates a greater difference in pathogenic property changes between the two locations.

Where the highlighted top properties are close to x=0, they are of similar importance at the two locations.

A shift away from y=0 indicates that effect sizes are greater at one location, even though rankings may be similar.

**Fig 6. Differences in property changes (delta properties) between locations plotted against rank change of those properties.** Effect size difference in the delta properties of two locations is on the Y-axis, and rank change of the delta properties is on the X-axis. The top properties (by effect size) for each location are coloured green and orange, while overlapping top properties are coloured red. A) Surface locations vs all interfaces. B) Interior locations vs all interfaces. C) Heteromeric vs homomeric interfaces. D) Heterologous vs isologous interfaces. E) DNA interfaces vs all protein interfaces.

indication of the scale of the distribution. Finally, the sum of absolute rank delta for the top 10 properties is calculated in the same way as the sum of all absolute rank deltas, but limited to only those top properties by effect size for each location (as in Table 1). This statistic indicates the scale of difference between the most important properties for pathogenesis at each location.

To address the question of whether interfaces resemble interior or surface locations the most, we plotted surface vs interface property changes (Fig 6A and S3 Table) and interior vs interface property changes (Fig 6B and S3 Table). The wider distribution on the x-axis in Fig 6A along with the greater absolute sum of all rank differences demonstrates that the property changes important for pathogenesis are more different between surface vs interface positions, and more similar between interior vs interface positions. Where the sum is smaller, and the distribution is correspondingly narrower, the two locations are more similar in terms of property changes between pathogenic and putatively benign variants. This is further corroborated by the fact that the top properties associated with pathogenesis in each location do not overlap in Fig 6A, whereas there is considerable overlap between the top properties from Table 1 in Fig 6B, indicating that the most important properties for pathogenesis are the same at both locations. This is further shown in S3 Table by limiting the sum of rank differences to only the top 10 properties, producing a much greater value for the surface vs interior chart. In addition, the entire plot in Fig 6B is shifted towards positive y-axis values, due to increased effect sizes of these properties at interior locations, despite similar relative importance at each location.

In heteromeric vs homomeric (isologous and heterologous combined) interfaces (Fig 6C and S3 Table), the distribution is very similar to that in Fig 6A. The limited overlap between top properties and low rank sum of the top properties in S3 Table indicate that more similar property changes are responsible for pathogenesis at these locations than between surface and interface amino acids. However, when we compare heterologous interfaces to isologous (Fig 6D and S3 Table) we find a much wider distribution characterised by the largest range of both effect size deltas and rank deltas, and complete separation of the top properties between the two locations. The top properties in this distribution are among those furthest from the origin and are thus those that differ in effect on pathogenesis the most between the two locations. This is also evident in S3 Table as the rank sum of the top properties was the largest observed. These were primarily properties related to secondary structure such as *relative preference value at C-cap* [56] and *normalized frequency of left-handed alpha-helix* [57]. This result could be biased by the presence of transmembrane helices in cyclic membrane proteins such as ion channels; however, exclusion of predicted membrane proteins yielded similar results (Fig J in S1 Text).

DNA interfaces also showed considerable differences when compared to protein-protein interfaces (Fig 6E and S3 Table). Although there was a single overlap between the top properties, they also represent some of those furthest from the origin and the second-largest rank sum of top properties, particularly *normalized frequency of N-terminal helix* [58] and *dependence of partition coefficient on ionic strength* [59]. Secondary structure once again appears to be a major factor. It is likely that the major differences between DNA interfaces and other interface types are those mutations that alter DNA specificity or abolish DNA binding.

## Performance of variant effect predictors on mutations from different locations and interface types

One of the areas of biology which has most benefited from the recent explosion in genome sequencing and increase in understanding of the mechanisms of pathogenesis, is that of variant effect prediction [60–62]. The effect of most coding variants on phenotype remains

completely unknown; therefore, these predictors are valuable in both clinical and research environments. The basis of almost all predictors is evolutionary conservation; however, the inclusion of additional features also often positively contributes towards accurate predictions. Given the varying properties of disease mutations at different locations and interface types we observed in the previous section, we believe that this could have implications for variant effect predictions.

In order to assess the current state of computational variant effect prediction by protein region, we obtained results from 29 computational predictors of variant effect from the dbNSFP database [63]. We then retrieved predictions for as many mutations in our datasets as possible and calculated the receiver operating characteristic (ROC) area under the curve (AUC) statistic for each interface type and location for each predictor. The calculations were repeated 1000 times by resampling the pathogenic and putatively benign datasets with replacement, and the average AUC statistic was taken. The results for four representative predictors (SIFT: an empirical method [64], DEOGEN2: a supervised method [65], PROVEAN: an unsupervised method [66] and REVEL: a metapredictor [67]) are shown in Fig 7, while full results are provided in Fig K in S1 Text.

Some interesting patterns emerge when comparing the performance of variant effect predictors across different protein locations. Out of surface, interior and interface mutations, surface mutants are most frequently the best predicted, while interfaces are the worst, a trend that matches their relative levels of pathogenic mutations. Between the different interface types, the outcomes of mutations at heterologous interfaces tend to be predicted particularly well, being better predicted than those at isologous interfaces for 28/29 predictors, and better than any interface locations for 18 predictors. In contrast, pathogenic mutations at DNA interfaces tended to be predicted quite poorly, being the worst predicted interface type for 21 predictors.

The consistency at which mutations at certain interface types are better predicted than others by variant effect predictors further demonstrates that the interface type has an impact on prediction efficiency across multiple methods. The consistency at which heterologous interfaces were better predicted than isologous interfaces was surprising. One possible explanation for this could be that complexes with heterologous interfaces tend to be more conserved and thus able to be better predicted than homodimers. The mechanisms of DNA and RNA binding, on the other hand are unlikely to be effectively captured by non-specialist variant effect predictors, as related transcription factors can bind to very different sequences.

Importantly, with this analysis, we are not particularly interested in the relative performance of different computational predictors. This is because many of the predictors will have been trained on some of the mutations in our dataset, and so their performance is almost certain to be overstated due to overfitting [68]. Thus, we cannot determine whether the top-performing predictors are genuinely better, or whether they have been trained more against mutations from our dataset. However, we hope that the effects of this bias should be smaller when comparing predictions for different groups made by the same computational method. For the sake of curiosity, we include a comparison between predictors at different locations and interface types using the area under the precision-recall curve (Fig L in S1 Text). It is interesting to note that the same few predictors tend to perform best across most groups, in particular M-CAP, DEOGEN2 and REVEL. In particular, it is interesting to note that DEOGEN2, the only predictor we are aware of that specifically uses interface information [65], ranks in the top two for core, support and rim regions, and in the top three for all interface types except RNA. While these results suggest it may be worthwhile trying these predictors if a mutation is known to occur at an interface residue, we are unable to state with confidence that they are genuinely better performing for interface mutations, based upon our results here.

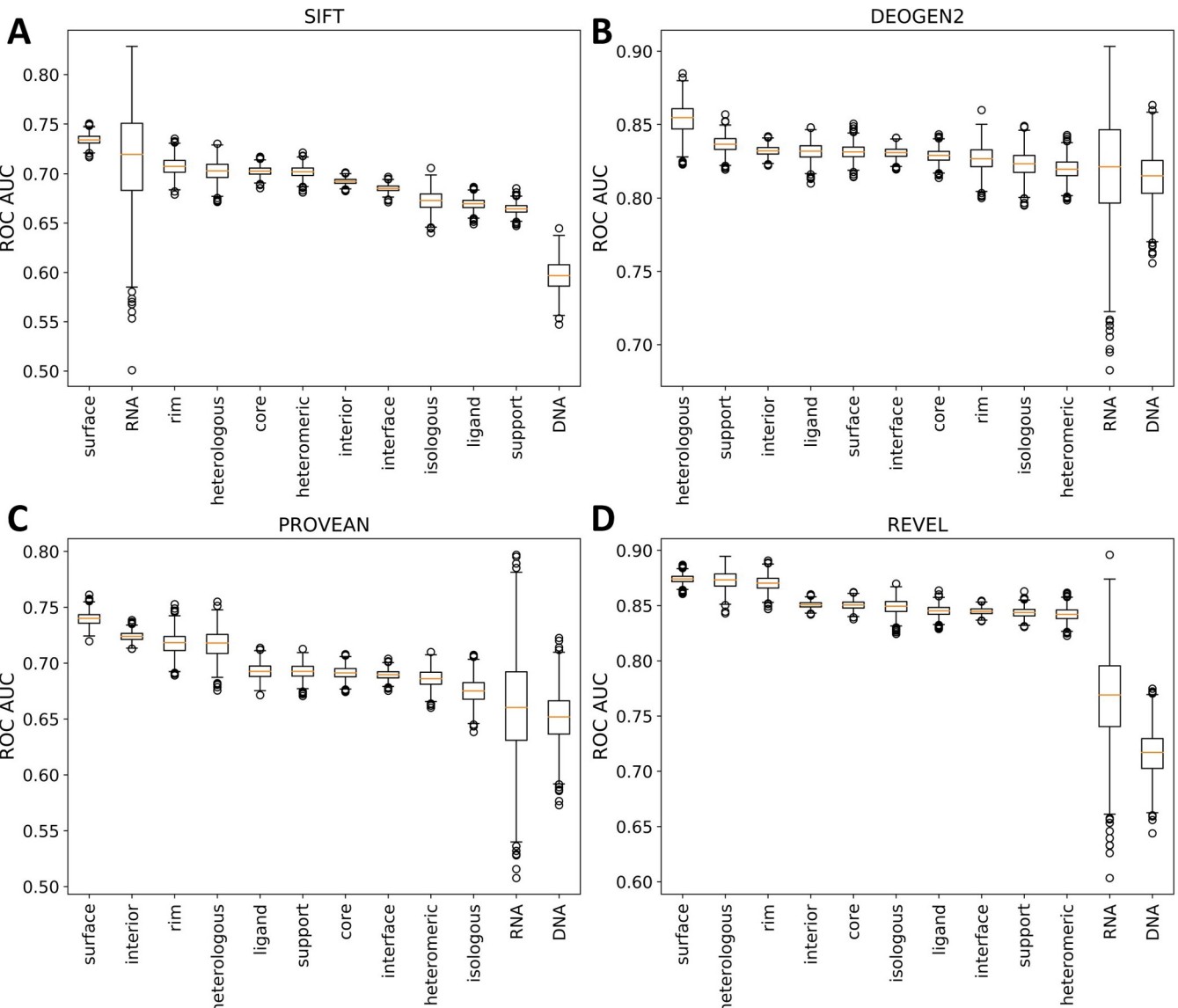

**Fig 7. Distribution of bootstrapped ROC AUC for predictions made by four representative variant effect predictors at different protein locations.** A) SIFT: an empirical predictor. B) DEOGEN2: a supervised machine learning method. C) PROVEAN: an unsupervised machine learning method. D) REVEL: a metapredictor (*i.e.* ensemble predictor). Charts for all predictors are available in Fig K in S1 Text.

## Conclusions

The goals of this study have been to assess the nature of differences in pathogenic and benign mutations in the context of protein interfaces, with a specific focus on findings that may inform or benefit future variant effect prediction methodology. The division of protein interfaces based on ligand type and on overall complex symmetry is an approach that has received little previous attention. Heterologous protein interfaces in particular demonstrated an increased enrichment of disease mutations relative to other protein-protein interface types. This was further demonstrated when cyclic, dihedral and helical complexes (all of which can contain heterologous interfaces) were also found to be enriched in pathogenic mutations. We ruled out ion-channel enrichment as a possibility for this observation. A more likely scenario

is that heterologous interfaces are a basis for the generation of larger complexes (through formation of fibrils, or larger closed cycles), and that slightly disruptive interface mutations have a larger effect on complex stability than on the stability of smaller complexes such as dimers. It is also likely that this effect increases with the number of mutant subunits forming the protein complex.

All locations within the protein demonstrated broadly similar patterns of tolerance towards specific amino acid substitutions. For example, regardless of location, mutations away from cysteine were damaging, as were mutations to proline. However, comparison of the relative enrichment ratios between locations pointed towards some interesting relationships. Mutations from cysteine on the protein surface were found to be relatively more damaging than at all other locations. Previous work has suggested that there is selection pressure to remove cysteines from the protein surface due to its high reactivity [69]; this implies that remaining surface cysteines have increased functional importance, thus explaining our observations. We also found evidence of differences between amino acids involved in pathogenic mutations within isologous and heterologous interfaces, and within DNA and protein interfaces.

Allostery has been of great interest to the scientific community of late, particularly in the field of drug discovery [70,71]. Our observation that sites distal to protein interfaces are enriched in pathogenic mutations in a distance-dependent manner is most likely a form of allostery. Indeed, allostery has been implicated as a mechanism involved in several disease mutations [47,48]. These findings also have implications for future variant annotations (both manual and through use of computational tools). While 3D protein structural information has been integrated into a number of variant effect predictors [72,73], interface proximity has not, so far, been used as a predictive feature, to the best of our knowledge. Our results indicate that inclusion of interface proximity as a feature may be beneficial in predicting variant pathogenicity.

Heterologous homomeric protein-protein interfaces have demonstrated some unexpected characteristics in comparison to other protein-protein interface types, even in comparison to isologous homomeric interfaces. This is most greatly reflected in the properties that exhibit the most change between pathogenic and benign mutations. While changes in hydrophobicity and related properties are among the most important for pathogenesis at almost every location, for heterologous interfaces (along with DNA interfaces), properties relating to secondary structure exhibit greater effects. While it is tempting to explain this as being due to transmembrane helices in membrane ion channels (of cyclic symmetry), removal of these proteins from the analysis does not alter the observation. It may be that secondary structure disruption is a more important factor at heterologous interfaces than altered hydrophobicity, which is an important observation for variant effect prediction at these sites. The case for protein-DNA interfaces is easier to explain, since DNA binding motifs often involve alpha helical motifs [74].

Computational variant effect predictors have been widely used for decades now. These programs are most commonly based on evolutionary conservation within a multiple sequence alignment. As a proxy metric for mutation tolerance, it performs well and is a major part of even the most modern predictors. Additional biochemical and structural features are also often included where they can aid performance. One aspect that is rarely considered when training a predictor, is the presence or the absence of a protein interface, or the type of protein interface. The consistent bias of all predictors in favour of heterologous interfaces and against DNA/RNA interfaces underscores the need for improvement in this area. Our results indicate several possible features which may be of benefit to future variant effect predictors including interface presence and type, distance to interface and location-specific property changes.

One potential pitfall of working with large datasets such as gnomAD and ClinVar is the potential for low-quality, contradictory or biased data to be included as part of the study. As

we have discussed, gnomAD is certain to contain some fraction of damaging variants associated with recessive, late-onset or incomplete penetrance disease, but this does not appear to strongly influence our results. However, for those variants annotated as pathogenic or likely pathogenic in ClinVar, it is possible that certain variant effect predictors were used when in the process of making those classifications. If these predictors took into account protein interface information, this could contribute to the enrichment of pathogenic ClinVar variants we see at interfaces. We believe this to be unlikely given the small numbers of predictors that make direct use of protein interface features. Similarly, knowledge that a mutation occurred in a functionally important interface could potentially have been used when stating in the literature that a new mutation was likely to be pathogenic, independent of the use of any computational predictor. Overall, we think the extent to which these factors could influence our results is likely to be very small. However, this potential for circularity is something that should be acknowledged in any analysis of pathogenic mutations.

Overall, our work has highlighted a number of interesting aspects of pathogenic mutations at protein interfaces. We hope that these findings will help improve variant effect predictor methodology and inspire further research in the area.

## Materials and methods

### ClinVar and gnomAD databases

To establish sets of pathogenic and benign mutations, we used the ClinVar [28], downloaded on 2020.08.17, and gnomAD v2.1.1 [29] databases. After filtering for missense variants, we selected only those variants in ClinVar labelled as "pathogenic" and "likely pathogenic". For the analysis of ClinVar benign variants, we considered those annotated as "benign" and "likely benign". Variants in gnomAD are aggregated from large-scale sequencing projects and exclude those individuals with severe paediatric disease. We refer to these variants as 'putatively benign', as some recessive and low penetrance variants no doubt remain in the dataset. We removed any variants annotated as "pathogenic" or "likely pathogenic" from the gnomAD set.

### PDB

Protein structures were downloaded from the Protein Data Bank (PDB) [75] on 2020.05.27. The first biological assembly was used to represent the quaternary structure of each protein, and symmetry assignments were taken directly from the PDB. We searched for those polypeptide chains with >90% sequence identity to a human protein over a region of at least 50 amino acid residues. Although this allows some closely related homologues to human structures to be included, we only considered those residues where the amino acid side chain in the structure, as well as the side chains of both adjacent residues, are the same as in the human sequence. For human protein residues that map to multiple chains, we selected a single chain sorting by highest resolution followed by largest structure. In the case of NMR ensembles, we selected only the first structure within the ensemble. Each variant is mapped to only a single position in a single structure, resulting in a fully non-redundant dataset. The full set of human ClinVar and gnomAD mutations mapped to protein structures is provided as a supplementary dataset.

### Ligand interfaces

Ligand interfaces were classified as 'cognate', 'non-cognate' or 'ambiguous' using the FireDB database of ligand significance [37]. As there were very few ligands in the ambiguous category, we investigated disease enrichments for only cognate and non-cognate ligands.

### Solvent accessible surface area

SASA is the area of the protein surface that is accessible to solvent molecules. SASA was calculated for each structure using AREAIMOL [76]. Residues were determined to be members of a protein interface if they experienced a change in SASA between the monomeric structure and the structure of the full PDB complex. SASA was then scaled [30] to determine the fraction of exposed residue. Residues with greater than or equal to 25% exposed surface area were classified as surface residues, while those with less were classified as interior. Those residues that changed from surface to interior upon complex formation were classified as interface core residues. Those that experienced a change in SASA but remained interior were classified as support residues and those that experienced a change but remained surface were classified as rim residues (Fig 1A).

### Odds ratios

Odds ratios, or relative enrichment values were calculated using the scipy.stats.fisher_exact() function from the python scipy package. For a pathogenic and putatively benign group within a subset of the data, this is equivalent to:

$$Odds\ Ratio = \left(\frac{disease\_subset}{disease\_non\_subset}\right) \Big/ \left(\frac{benign\_subset}{benign\_non\_subset}\right)$$

Confidence intervals (95%) for odds ratios were determined by:

$$CI = \hat{e}(\ln(OR) \pm 1.96\sqrt{(\frac{1}{disease\_subset} + \frac{1}{disease\_non\_subset} + \frac{1}{neutral\_subset} + \frac{1}{neutral\_non\_subset})})$$

where CI is the upper/lower confidence interval and OR is the odds ratio.

### 3D complex residue distance calculations

Distances between interior/surface mutations and interfaces were calculated using atomic coordinates in PDB files. The coordinates of alpha-carbon atoms were used to represent the location of all residues. Rim residues were removed from all interfaces. For each PDB structure, the centroid of each interface was calculated as the mean of the x, y and z coordinates of every amino acid in the interface. The distance between each interior/surface mutation and interface was calculated using:

$$distance = \sqrt{(x_1 - x_2)^2 + (y_1 - y_2)^2 + (z_1 - z_2)^2}$$

for each interface within the PDB chain. Where multiple interfaces were present in the structure, then the distance to the closest interface was used to represent the proximity of the mutation. To calculate linear sequence proximity to an interface, the absolute sequence distance to the closest interface residue was used. For all distance metrics, only the closest interface by that metric was considered, thus each variant only appears once in our data even when multiple interfaces are close to the variant.

### Variant effect predictors

Variant effect predictions were obtained from the dbNSFP database [63], with the exception of SIFT which was run locally using the Uniref90 database [77] to generate alignments.

### ROC curves and precision recall curves

Receiver operating characteristic (ROC) area under the curve (AUC) statistics were generated using the sklearn.metrics.roc_auc_score() function of the sklearn python package. Several variant effect predictors used inverse metrics to score variants (lower, rather than higher score representing greater probability of damaging variants). To resolve this issue, we deducted the score from 1.0 for all predictors with an AUC less than 0.5. Precision recall AUC was calculated using the sklearn.metrics.precision_recall_curve() and sklearn.metrics.auc() functions of the sklearn python package.

### AAindex

Amino acid properties and substitution matrices were obtained from the AAindex database [53]. Properties without values for all 20 standard amino acids were removed from the analysis.

### Property deltas

Differences in properties were calculated for all individual amino acid substitutions. A two-tailed student's t-test was then performed to determine significance of the difference between the mean of pathogenic and putatively benign mutation properties at different locations. Hedges' g was used to calculate effect size between the same.

As only small effect sizes were initially observed, we decided to remove directionality from the property delta values by taking the absolute differences instead. These new 'directionless' property deltas represent only the scale of the difference between the two property values.

### Supporting information

**S1 Text. Fig A in S1 Text: Enrichment of pathogenic variants at cognate and non-cognate ligand interfaces.** Error bars represent 95% confidence intervals. Enrichment values shown are relative to the full dataset. Detailed relative enrichment data is available in S1 Table. **Fig B in S1 Text: The impact of 'benign' dataset on the enrichment of pathogenic mutations.** Relative enrichment of pathogenic mutations using common ($\geq$1%) gnomAD variants, rare (<1%) gnomAD variants and ClinVar benign and likely benign variants in: A) different locations (surface, interior and interface regions); B) different interface regions (core, support and rim); and C) different protein interface types. All error bars represent 95% confidence intervals. Detailed relative enrichment data is available in S1 Table. **Fig C in S1 Text: Enrichment of pathogenic variants at different protein locations and interface types when excluding predicted membrane proteins.** A) Enrichment of pathogenic variants in surface, interior and interface locations relative to all data. B) Enrichment of pathogenic variants in different interface types relative to all data. Total numbers of pathogenic and putatively benign variants are shown below the plots. Error bars represent 95% confidence intervals. Detailed relative enrichment data is available in S1 Table. **Fig D in S1 Text: The impact of protein biological function on the enrichment of pathogenic mutations.** A) The enrichment of pathogenic mutations at different locations and interface types for proteins annotated as having/lacking catalytic activity in Uniprot. B) The enrichment of pathogenic mutations at different locations and interface types for proteins annotated as being/not being involved in transcriptional regulation in Uniprot. C) The enrichment of pathogenic mutations at different locations and interface types for proteins annotated as having/lacking transporter activity in Uniprot. All error bars represent 95% confidence intervals. Detailed relative enrichment data is available in S1 Table. **Fig E in S1 Text: The impact of interface size on the enrichment of pathogenic**

**variants.** Enrichment of pathogenic variants within interface residues from interfaces of different sizes. Each interface type was split into three groups of small, medium and large interfaces, containing equal numbers of variants. A) Homomeric isologous; B) homomeric heterologous; C) heteromeric; D) ligand; E) DNA; and F) RNA interface residues. Error bars represent 95% confidence intervals. Detailed relative enrichment data is available in S1 Table. **Fig F in S1 Text: The impact of protein length and variant number on the enrichment of pathogenic mutations.** A) The enrichment of pathogenic mutations at different locations and interface types for proteins of different amino acid chain lengths ($<$ = 500 residues, 500–1000 residues, $>$1000 residues). B) The enrichment of pathogenic mutations at different locations and interface types for proteins with different numbers of variants in our datasets (combined ClinVar and gnomAD). All error bars represent 95% confidence intervals. Detailed relative enrichment data is available in S1 Table. **Fig G in S1 Text: Odds ratio of disease mutations at increasing distance from the nearest interface.** A) Using the minimum distance to the nearest interface residue. B) Using the mean distance to all interface residues. 95% confidence intervals are shown in grey, a univariate spline (blue) has been fit to the data. Charts are shown for surface and interior residues together and individually. **Fig H in S1 Text: Enrichment of pathogenic variants involving different amino acid residues for different types of interfaces.** The odds ratio of pathogenic mutations associated with specific amino acid mutations at each interface type as well as core and support regions. Blue bars represent mutations *from* a specific amino acid, while orange bars represent mutations *to* a specific amino acid. Error bars represent 95% confidence intervals. Detailed relative enrichment data is available in S1 Table. **Fig I in S1 Text: Relative enrichment of pathogenic variants for cysteine and proline mutants.** A) Enrichment of pathogenic variants for all mutations possible via a single nucleotide change from wild-type cysteine at different protein locations. B) Enrichment of pathogenic variants for all wild-type amino acids that could mutate to proline via a single nucleotide change. All error bars represent 95% confidence intervals. Detailed relative enrichment data is available in S1 Table. **Fig J in S1 Text: Difference in property changes between isologous and heterologous interfaces plotted against rank change of those properties with all predicted membrane proteins excluded.** Effect size difference in the delta properties of two locations is on the Y-axis, and rank change of the delta properties is on the X-axis. The top properties (by effect size) for each location are coloured green and orange. **Fig K in S1 Text: Distribution of bootstrapped ROC AUC for predictions made by different computational methods.** Pathogenic and putatively benign datasets were independently re-sampled 1000 times with replacement. Only mutations with all 29 predictions were included in this analysis. **Fig L in S1 Text: Distribution of bootstrapped precision recall AUCs for variant effect predictions at different protein locations and interface types.** Pathogenic and putatively benign datasets were independently re-sampled 1000 times with replacement. Only mutations with all 29 predictions were included in this analysis.
(DOCX)

**S1 Table. Sample numbers, odds ratios, confidence intervals and p-values for all enrichment analyses.**
(XLSX)

**S2 Table. All AAindex amino acid property changes by effect size per location.**
(XLSX)

**S3 Table. Summary statistics for distributions plotted in Fig 6.**
(XLSX)

## Author Contributions

**Conceptualization:** Joseph A. Marsh.

**Formal analysis:** Benjamin J. Livesey.

**Supervision:** Joseph A. Marsh.

**Writing – original draft:** Benjamin J. Livesey.

**Writing – review & editing:** Joseph A. Marsh.

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
