## [Decision Letter · Decision Letter 0]

11 Oct 2021

Dear Marsh,

Thank you very much for submitting your manuscript "The properties of human disease mutations at protein interfaces" for consideration at PLOS Computational Biology.

As with all papers reviewed by the journal, your manuscript was reviewed by members of the editorial board and by several independent reviewers. In light of the reviews (below this email), we would like to invite the resubmission of a significantly-revised version that takes into account the reviewers' comments.

We cannot make any decision about publication until we have seen the revised manuscript and your response to the reviewers' comments. Your revised manuscript is also likely to be sent to reviewers for further evaluation.

Sincerely,

Petras J Kundrotas

Guest Editor

PLOS Computational Biology

Arne Elofsson

Deputy Editor

PLOS Computational Biology

Reviewer's Responses to Questions

**Comments to the Authors:**

Reviewer #1: Review upload as attachment

Reviewer #2: This paper studies the enrichment of pathogenic variants according to their protein locations or interfaces. The differences in terms of protein interfaces may influence the potential pathogenicity and can be included as a factor in variant effect prediction. The study in general would be useful for exploring the effect of protein interfaces on the pathogenicity of variants. The paper is well written and easy to understand. I list my comments below.

1. There might be some kind of circular logic in the analysis, which is related to the difficulty of defining pathogenic variants. The paper uses ClinVar data to define pathogenicity, which are likely influenced/predicted by variant effect prediction tools, and many tools might have used features correlated with the protein interface to train their classifiers. Therefore, the protein interfaces might have already been considered as a factor explicitly/implicitly to define pathogenicity. We need to keep this potential circular logic in mind when interpreting the analysis results. The authors might discuss this in the discussion or conclusion section.

2. The analysis pools all genes/proteins together and do the enrichment analysis using Fisher’s exact test. We need to be cautious about potential confounding factors or heterogeneity ignored when pooling all genes together. To be more convincing, it would be helpful to stratify the genes/proteins into different groups and do the analysis within each group to make sure the patterns observed can still hold. Here are some suggested stratifications: 1) stratify genes into bins of different gene length; 2) stratify genes into bins of different number of variants; 3) stratify genes by whether they are transcription factors or not.

3. Figure 6 and related explanation in the manuscript is not easy to understand, not intuitive to show the differences between different location types. We don’t know how statistically significant of these differences as described in the main text. For example, a “narrow” distribution is hard to define.

4. In the section: Performance of variant effect predictors on interface mutations, the ROC statistics is used. For data sets with very unbalanced positive & negative samples, the precision-recall curve might be more informative. Can the authors add the comparisons using the precision-recall AUC?

5. Figure 5 shows high enrichment for mutations away from cysteine and mutations to proline. Strictly speaking, a mutation is defined by both the “from” and “to”. Can the author also show the enrichment results of all mutations from cysteine? Similarly, the enrichment results of all mutations to the damaging proline? Is there a specific dominating mutation pattern or there is an almost even enrichment for all patterns?

6. When collecting variants from gnomAD, is there an allele frequency threshold? Or matching the variant allele frequency in ClinVar?

7. Figure 5 legend: It should be odds ration instead of log odds ratio

8. The following is not correctly referenced: Vitkup et al, 2003

Reviewer #3: Title; The properties of human disease mutations at protein interfaces.

Comments;

I would like to suggest authors to include most deleterious mutations (Experimentally proved) and the least deleterious mutations in the current data set.

Any insight on the mutation(s) altering interface (protein-RNA, protein-protein, Protein-DNA and protein-ligand ect) simultaneously.

An explanation required on location (near, far ,very far from the interface) of the mutation site and its disease association.

Does the author point out any particular mutation /type affecting various interactions or the frequent type of mutation?

It will be good a information if authors incorporate frequency of mutation and the disease association. Second information on the nature of amino acid mutation and the disease association.

**Have the authors made all data and (if applicable) computational code underlying the findings in their manuscript fully available?**

Reviewer #1: Yes

Reviewer #2: Yes

Reviewer #3: Yes

PLOS authors have the option to publish the peer review history of their article (what does this mean?). If published, this will include your full peer review and any attached files.

Reviewer #1: No

Reviewer #2: **Yes: **Wenan Chen

Reviewer #3: No
---

## [Decision Letter · Decision Letter 1]

7 Jan 2022

Dear Marsh,

Thank you very much for submitting your manuscript "The properties of human disease mutations at protein interfaces" for consideration at PLOS Computational Biology. As with all papers reviewed by the journal, your manuscript was reviewed by members of the editorial board and by several independent reviewers. The reviewers appreciated the attention to an important topic. Based on the reviews, we are likely to accept this manuscript for publication, providing that you modify the manuscript according to the review recommendations.

Sincerely,

Petras J Kundrotas

Guest Editor

PLOS Computational Biology

Arne Elofsson

Deputy Editor

PLOS Computational Biology

[LINK]

Reviewer's Responses to Questions

**Comments to the Authors:**

Reviewer #1: The authors have addressed all comments and the revised manuscript can be accepted for publication

Reviewer #2: The authors addressed my questions well except the following one:

It is not the precision score metric I suggested but the area under the curve (AUC) of the precision-recall curve. The ROC curve uses the sensitivity and specificity as x and y axis, the precision-recall curve uses the precision and the sensitivity as the x and y axis. We can calculate the AUC of either the precision-recall curve or the ROC curve, as long as positive and negative samples are available and a score is defined for each sample. The should both work for the purpose of comparing the performance stratified by the protein locations and interface types. Usually the AUC of the precision-recall curve gives us another view of the performance especially when the positive-negative ratio is far away from 1. The authors can check the REVEL paper where both AUC of ROC curve and precision-recall curve were calculated: https://www.ncbi.nlm.nih.gov/pmc/articles/PMC5065685/

Reviewer #3: ./

**Have the authors made all data and (if applicable) computational code underlying the findings in their manuscript fully available?**

Reviewer #1: Yes

Reviewer #2: Yes

Reviewer #3: Yes

PLOS authors have the option to publish the peer review history of their article (what does this mean?). If published, this will include your full peer review and any attached files.

Reviewer #1: No

Reviewer #2: **Yes: **Wenan Chen

Reviewer #3: **Yes: **Dr. Rituraj Purohit

Figure Files:

Data Requirements:

Reproducibility:

References:

---

## [Decision Letter · Decision Letter 2]

24 Jan 2022

Dear Marsh,

We are pleased to inform you that your manuscript 'The properties of human disease mutations at protein interfaces' has been provisionally accepted for publication in PLOS Computational Biology.

Best regards,

Petras J Kundrotas

Guest Editor

PLOS Computational Biology

Arne Elofsson

Deputy Editor

PLOS Computational Biology

Reviewer's Responses to Questions

**Comments to the Authors:**

Reviewer #2: Good, no further questions. Thanks.

**Have the authors made all data and (if applicable) computational code underlying the findings in their manuscript fully available?**

Reviewer #2: Yes

PLOS authors have the option to publish the peer review history of their article (what does this mean?). If published, this will include your full peer review and any attached files.

Reviewer #2: **Yes: **Wenan Chen

---

## [Editor Report · Acceptance letter]

1 Feb 2022

PCOMPBIOL-D-21-01554R2 

The properties of human disease mutations at protein interfaces

Dear Dr Marsh,

I am pleased to inform you that your manuscript has been formally accepted for publication in PLOS Computational Biology. Your manuscript is now with our production department and you will be notified of the publication date in due course.

With kind regards,

Orsolya Voros
